# Aligning with Logic: Measuring, Evaluating and Improving Logical Preference Consistency in Large Language Models

Yinhong Liu [1]   Zhijiang Guo [1]   Tianya Liang [1]   Ehsan Shareghi [2]   Ivan Vulić [1]   Nigel Collier [1]

## Abstract

Large Language Models (LLMs) are expected to be predictable and trustworthy to support reliable decision-making systems. Yet current LLMs often show inconsistencies in their judgments. In this work, we examine *logical preference consistency* as a foundational requirement for building more dependable LLM systems, ensuring stable and coherent decision-making while minimizing erratic or contradictory outputs. To quantify the logical preference consistency, we propose a universal evaluation framework based on three fundamental properties: *transitivity*, *commutativity* and *negation invariance*. Through extensive experimentation across diverse LLMs, we demonstrate that these properties serve as strong indicators of judgment robustness. Furthermore, we introduce a data refinement and augmentation technique, REPAIR, that enhances logical consistency while maintaining alignment with human preferences. Finally, we show that improving consistency leads to better performance in LLM-driven logic-based algorithms, reinforcing stability and coherence in decision-making systems. Code is available at https://github.com/williamLyh/REPAIR

## 1. Introduction

Recent research in Large Language Models (LLMs; Brown et al. 2020; OpenAI 2023; Anil et al. 2023a;b) has achieved substantial progress on various tasks, enabling them to generate responses that better support their decision-making and problem-solving (Dai et al., 2024). However, key challenges still exist regarding the reliability and trustworthiness of LLMs. Issues such as hallucination (Zhang et al., 2023), bias (Gallegos et al., 2024), and inconsistencies in reasoning (Huang & Chang, 2023) continue to affect their credibility.

These limitations hinder the full deployment of LLMs, particularly in professional and high-stakes applications.[3]

The foundation of a reliable and trustworthy system is the **consistency** of its predictions. A consistent system produces explainable and tractable decisions, enhancing its dependability and reliability. In this work, we focus on a key form of consistency in LLMs: **logical preference consistency**, which is critical for applications requiring structured reasoning and coherent decision-making (Creswell et al., 2023; Hamon et al., 2020). Logical inconsistencies can lead to unreliable conclusions (Restall, 2002) and even paradoxes (Hyde, 2011), posing significant risks in domains that demand rigorous logical judgment, such as temporal or spatial reasoning (Mostafazadeh et al., 2016b), optimization (Guo et al., 2024), and automated decision systems.

This paper examines three key aspects of logical preference consistency in LLMs: **transitivity**, **commutativity**, and **negation invariance**. We propose a universal framework for quantifying these consistency properties, applicable to any number of items, and adaptable across various domains. Our evaluations reveal a strong correlation between consistency and LLMs' judgment robustness, suggesting that logical preference consistency serves as a useful proxy for preference reliability. To enhance the logical preference consistency of LLMs, we propose REPAIR, a framework that refines noisy pairwise comparisons using rank aggregation and extrapolates additional comparisons logically. Models trained with this approach achieve better internal consistency without sacrificing alignment with human preferences. Moreover, in logic-dependent tasks, these models outperform less consistent ones, demonstrating improved efficiency in sorting-based ranking algorithms (Liu et al., 2024) reliant on logical coherence.

In sum, our contributions are as follows: **1)** We highlight the importance of logical preference consistency—alongside human alignment—in aligning LLM preferences. We define mathematical formulations for quantifying/measuring three key consistency properties: transitivity, commutativity and

---

[1]University of Cambridge [2]Monash University. Correspondence to: Yinhong Liu <yl535@cam.ac.uk>.

*Proceedings of the $42^{nd}$ International Conference on Machine Learning*, Vancouver, Canada. PMLR 267, 2025. Copyright 2025 by the author(s).

---

[3]For instance, in the fields of behavioral economics and psychology systems are traditionally evaluated on two key dimensions: *validity* and *reliability* (Schmidt et al., 2000; Guion, 2004; Miller et al., 2023).

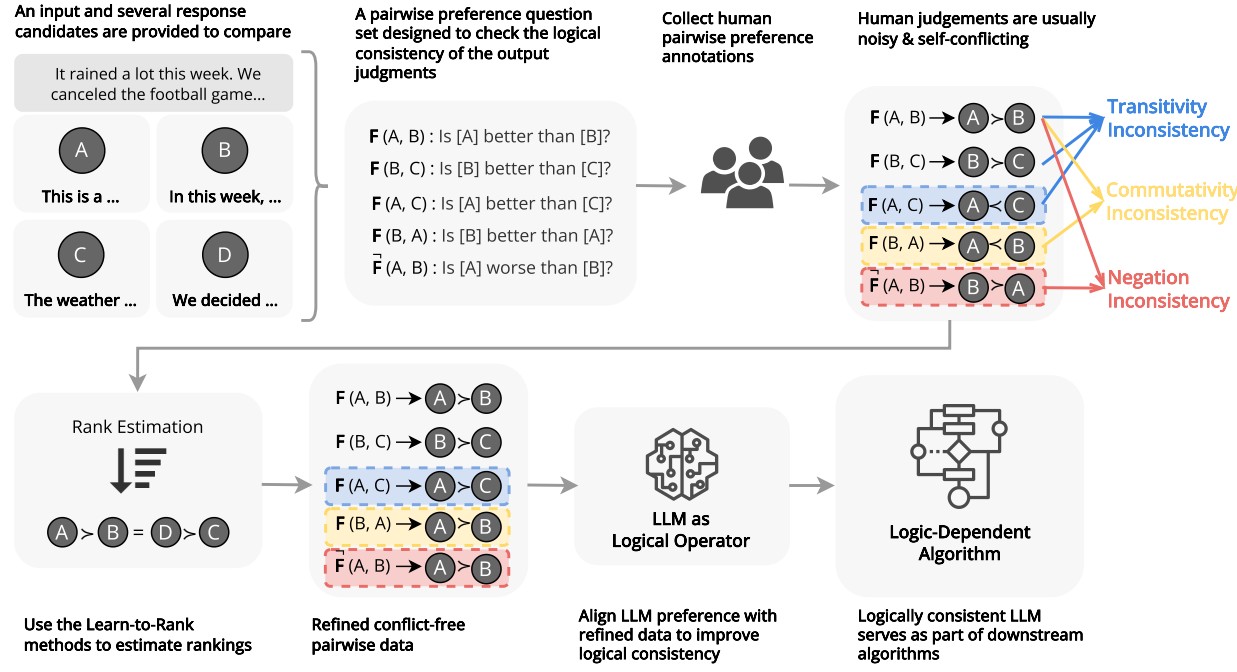

Figure 1: Three types of logical inconsistencies are observed in real-world pairwise annotations (top row): Transitivity, Commutativity, and Negation Invariance. By refining the data for self-consistency using rank estimation, we can train LLMs with enhanced logical consistency, improving their performance in logic-dependent algorithms (bottom row).

negation invariance. **2)** We conduct extensive experiments to evaluate logical preference consistency across state-of-the-art LLMs and analyze its correlation with model reliability. **3)** We propose a data refinement and augmentation method for instruction-tuning that enhances logical preference consistency while maintaining alignment with human preferences and **4)** we demonstrate that improving logical consistency enhances LLM performance in logic-dependent algorithms where LLMs serve as logical operators. We release our source code and the refined dataset at ANONYMOUS.

## 2. Measuring Logical Consistency

We evaluate the logical preference consistency of LLMs by assessing their ability to predict logically consistent relations among a *set* of *items*. These items could represent diverse *entities* or *events* with a uniform *relation* between them; such a relation might be (i) comparing the preference among response candidates to a query or (ii) determining the causal order of shuffled events, among other possibilities. This evaluation is grounded in relational algebra and order theory (Abiteboul et al., 1995; Imieliński & Lipski Jr, 1984), with the goal of assessing whether the model maintains coherent and self-consistent judgments across its predictions.

To formalize this concept, we define the *logical preference*

*consistency evaluation process* by treating an LLM as an operator function that compares pairs of items and outputs a decision on the relation between the items. Let $X = \{x_1, x_2, \ldots, x_N\}$ represent a set of items, and we define a relation (e.g., comparison) function $F : X \times X \to R$, which compares two items, such as $(x_i, x_j)$, and assigns a relational decision $F(x_i, x_j) = r$, where $r \in R$ denotes the directional relation between $x_i$ and $x_j$. For simplicity, we consider $R$ to be a binary relation set, $R = \{r_{ij}, r_{ji}\}$, where $r_{ij}$ represents a preferential relation $x_i \succ x_j$ (i.e., item $x_i$ is preferred over item $x_j$), and $r_{ji}$ indicates the reverse preference $x_j \succ x_i$.

In evaluating logical consistency, we focus on whether the function $F$ adheres to the following key properties over the item set $X$: *1) transitivity*, *2) commutativity*, and *3) negation invariance*, as demonstrated in Figure 1. Transitivity ensures that the LLM's predictions and judgements are internally coherent and do not suffer from logical contradictions within a given context. Commutativity tests whether the model's decisions are invariant to the order in which items are compared. The negation invariance checks whether the model maintains consistent understanding when dealing with relational negations. By systematically applying these tests, we are able to assess the extent to which the model's judgments conform to logically consistent behavior, providing a

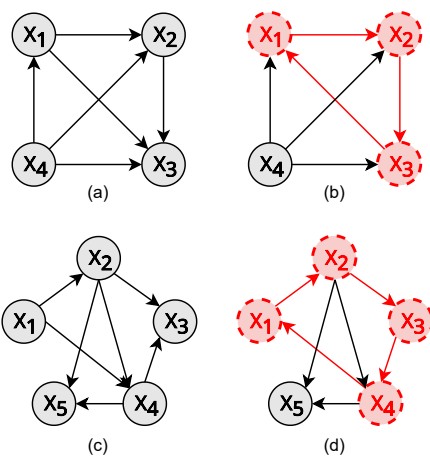

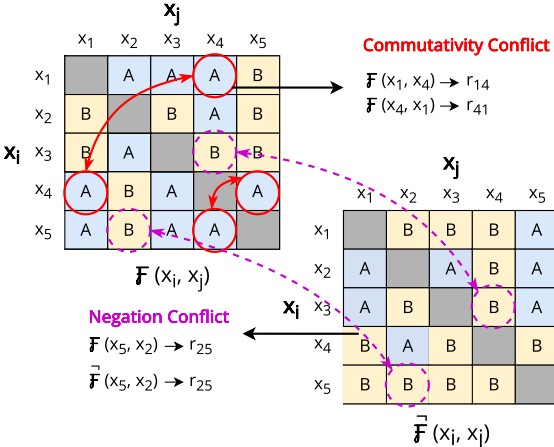

Figure 2: Example of relation graphs illustrating transitivity, where items are represented as nodes, and directed edges indicate pairwise preferential relations. Red dashed cycles in the graph highlight violations of transitivity. The cycle in (d), spanning 4 items, cannot be captured by $s_{tran}(3)$. The $s_{tran}$ metric can be applied to partial relation graphs, as shown in (c) and (d).

Figure 3: Examples illustrating violations of commutativity and negation invariance. Each entry of the two preference matrices represents predicted judgments of $x_i \succ x_j$ and $x_i \prec x_j$, labelled with A and B respectively. The top-left matrix is based on the original relation, while the bottom-right matrix reflects the negated relation. Linked red cycles highlight non-commutative pairs, and linked dashed purple cycles indicate negation inconsistencies.

quantitative proxy measure of its decision reliability.

### 2.1. Measuring Transitivity

Grounded in our problem setup and definitions above, transitivity implies that if a model predicts $A \succ B$ and $B \succ C$, it must also predict $A \succ C$. Ensuring transitive predictions is essential for a coherent global understanding of item relationships, preventing contradictions that could undermine reliability in decision-making or ranking tasks (Liu et al., 2024; Li et al., 2019; Qin et al., 2024).

We define that the function $F$ is fully transitive if it does not predict any intransitive relation within the set $X$. This means that if $F(x_i, x_j) = r_{ij}$ and $F(x_j, x_k) = r_{jk}$, then $F(x_i, x_k) = r_{ik}$ must hold for all $i, j, k \in X$. This can be visualized in Figure 2, where we represent the pairwise relations by $F$ as a relation graph. If $F$ is transitive over $X$, the corresponding relation graph should be a Directed Acyclic Graph (DAG). [4] Consequently, to determine if $F$ is fully transitive over the item set $X$, we only have to verify whether the predicted relation graph contains any cycle. We show how to construct the relation graph from the judgements of the LLM operator function $F$, and the algorithm to check whether a graph contains cycles in Appendix §A.

We introduce $s_{tran}(K)$ to quantify transitivity over an ar-

bitrary number of items. LLMs often struggle to maintain perfect/full transitivity, especially as the number of items increases. The proposed metric $s_{tran}(K)$ captures the *degree of transitivity* across subsets of $K$ sampled items, where $3 \leq K \leq |X|$. The metric is defined as:

$$s_{tran}(K) = \frac{1}{M} \sum_{i=1}^{M} \mathbb{1}_{\text{acyclic}}(S_i^K). \quad (1)$$

Here, $S_i^K$ represents a randomly sampled sub-graph of size $K$, and the indicator function $\mathbb{1}_{acyclic}$ returns 1 if the sub-graph $S_i^K$ contains no cycles (i.e., is transitive), and 0 if otherwise. $M$ denotes the total number of the sampled sub-graphs. As the size of the item set increases, the number of possible sub-graph combinations grows exponentially. To manage this complexity, we cap the number of samples to 1,000 sub-graphs to estimate transitivity for larger sets. The choice of $M$ is supported by theoretical evidence explained in Appendix §I. Therefore, the metric ranges from 0 to 1, where 1 represents nearly perfect transitivity.

Maintaining transitivity becomes increasingly difficult as the size of the subset grows. For any set of $K$ items, there are $2^K$ possible combinations of pairwise relationships, but only $K!$ of these can form a transitive ranking. As a result, the degree of transitivity tends to decrease with larger sub-graph sizes. This means that for the same LLM, $s_{tran}(K)$ typically decreases as $K$ increases, reflecting that preserving consistent rankings over larger sets of items is increasingly

---

[4] A DAG implies that *no cycles* exist in the relation graph. We assume the comparison $F$ is irreflexive, explained in Appendix §J

challenging. Since $s_{tran}(K)$ measures transitivity for a fixed subset size, it allows for fair transitivity comparisons between item sets of different sizes.

## 2.2. Measuring Commutativity

Commutativity refers to the logical property that ensures the model's judgments remain consistent when the order of comparison between two items is reversed. Prior studies have shown that LLMs are susceptible to permutation bias, also referred to as positional bias (Wang et al., 2024b; Liusie et al., 2024a). To measure the degree of commutativity, we propose a metric $s_{comm}$, which evaluates whether the model's judgment changes when the order of the items is swapped in the prompt. Specifically, it is defined as follows:

$$s_{comm} = \frac{2}{|X|(|X|-1)} \sum_{i<j} \mathbb{1}(F(x_i, x_j) = F(x_j, x_i)).$$
(2)

Here, $F(x_i, x_j)$ represents the model's judgment when comparing items $x_i$ and $x_j$. The indicator function $\mathbb{1}$ returns 1 if the model's judgment remains consistent when the order of the items is reversed, i.e., $F(x_i, x_j) = F(x_j, x_i)$, and 0 otherwise. We visualize this comparison in Figure 3. The normalization term ensures that $s_{comm}$ is averaged across all pairwise combinations of the items in set $X$. As a result, the metric ranges from 0 to 1, with 1 indicating perfect commutativity, meaning that the model is completely robust to the order of item comparisons.

## 2.3. Measuring Negation Invariance

Negation invariance tests whether the model maintains consistency when confronted with the negation or inversion of a relational statement. Inconsistencies indicate a failure to correctly understand and apply the complementary nature of relations. Previous work suggested that LLMs struggle to automatically infer appropriate inverse relationships when acquiring knowledge (Allen-Zhu & Li, 2023; Berglund et al., 2024). To quantify negation invariance, we propose the metric $s_{neg}$, which examines if the model can correctly reverse its judgement when prompted with a negated relationship between items. The metric is defined as below:

$$s_{neg} = \frac{1}{|X|(|X|-1)} \sum_{\substack{0<i,j\leq|X| \\ i\neq j}} \mathbb{1}(\bar{F}(x_i, x_j) = \neg F(x_i, x_j)).$$
(3)

In this formulation, $\neg F(x_i, x_j)$ represents the negation of the original relation (e.g., reversing a preference or relational direction). $\bar{F}(x_i, x_j)$ refers to the model's judgment when explicitly prompted with the negated relation. The indicator function returns 1 if the model's response to the negated relation matches the expected negated judgment (i.e., $\bar{F}(x_i, x_j) = \neg F(x_i, x_j)$), and 0 otherwise. We also

visualize this comparison in Figure 3. The normalization factor averages across all pairwise permutations in set $X$, ensuring that $s_{neg}$ ranges from 0 to 1. The maximum score of 1 indicates perfect negation invariance, where the model consistently handles negated relations.

# 3. Evaluating Logical Consistency of LLMs

After defining the measures, we proceed to evaluate LLMs' judgments from the consistency angle on three representative tasks, each reflecting different levels of subjectivity.

## 3.1. Evaluation Setup

**Tasks and Datasets.** We employ three representative tasks to evaluate LLMs' logical consistency. The first task, *abstractive summarization evaluation*, uses the SummEval dataset (Fabbri et al., 2021) and focuses on the model's ability to make preference judgments between summaries, particularly regarding the coherence aspect. The second task, *document reranking*, leverages the NovelEval dataset (Sun et al., 2023), where LLMs assess the relevance of retrieved documents in response to queries. The final task, *temporal event ordering*, uses the CaTeRS dataset (Mostafazadeh et al., 2016b) and tests the model's capability to judge temporal and causal relationships between events, critical for maintaining consistency in narrative understanding. Further task and dataset details are available in Appendix §B.

**Metrics and Reliability Measurement.** In Appendix §G, we show metric computation details and the prompt templates used. We compute the logical consistency metrics at the instance level and report averages across the test sets for each task. In addition, we report the human agreement rate (abbreviated as H.) by calculating the pairwise judgement accuracy between judgements made by LLMs and the provided human annotations. It serves as a reference for how closely the model's judgements align with human judgements. For the measurement of LLMs' reliability, we perform Monte Carlo Sampling of Chain-of-Thought (Wei et al., 2022) reasoning outputs, using a temperature of 0.7. Self-agreement is defined as the percentage of outputs that agree with the majority judgment across multiple samples. This measurement ranges from 0.5 to 1, with higher values indicating greater stability in the model's judgments.

## 3.2. Results and Analysis

**Performance of Different (Families of) Models.** As shown in Table 1, recent LLMs like Gemma2-9B and Phi-3-medium demonstrate stronger overall consistency compared to earlier models. In particular, models such as Deepseek-chat-v2, Phi-3-medium, and Gemma-2-9B perform well across all three evaluated consistency dimensions. However, it is important to note that strong performance in one aspect

Table 1: Logical consistency evaluation results. We report human accuracy (H.), transitivity over 5 items ($s_{tran}(5)$), commutativity ($s_{comm}$) and negation invariance ($s_{neg}$), all measured in accuracy.

| Models | SummEval (Coh) | | | | NovelEval | | | | CaTeRS | | | |
|---|---|---|---|---|---|---|---|---|---|---|---|---|
| | H. | $s_{tran}(5)$ | $s_{comm}$ | $s_{neg}$ | H. | $s_{tran}(5)$ | $s_{comm}$ | $s_{neg}$ | H. | $s_{tran}(5)$ | $s_{comm}$ | $s_{neg}$ |
| Direct Judgements | | | | | | | | | | | | |
| Llama-2-7B | 57.5 | 88.3 | 57.5 | 66.2 | 57.5 | 68.1 | 57.5 | 78.1 | 61.9 | 88.4 | 56.0 | 49.7 |
| Llama-2-13B | 58.3 | 86.6 | 59.3 | 84.0 | 58.3 | 88.2 | 62.9 | 76.9 | 65.6 | 95.3 | 70.8 | 54.5 |
| Llama-3-8B | 67.8 | 91.0 | 76.1 | 48.9 | 60.6 | 73.0 | 73.3 | 79.1 | 73.1 | 88.2 | 79.9 | 63.3 |
| Mistral-7B | 63.6 | 95.1 | 59.9 | 51.2 | 60.1 | 90.5 | 68.0 | 82.1 | 70.2 | 95.9 | 73.9 | 76.9 |
| Zephyr-7B-beta | 61.3 | 87.8 | 52.8 | 74.1 | 60.5 | 91.5 | 63.8 | 86.5 | 70.6 | 93.5 | 77.8 | 80.8 |
| Phi-3-mini | 65.6 | 92.8 | 66.9 | 75.1 | 59.7 | 92.5 | 55.5 | 39.4 | 60.2 | **97.2** | 85.5 | 73.3 |
| Phi-3-medium | 68.8 | **96.2** | 71.0 | 78.1 | 62.7 | 93.7 | 76.3 | 77.6 | 67.2 | 93.4 | 87.9 | 84.1 |
| Gemma-2-9B | **73.8** | 94.8 | **78.1** | 78.2 | 63.6 | 89.9 | 85.3 | 88.2 | 72.4 | 96.0 | 86.5 | 66.9 |
| GPT-3.5-0125 | 66.3 | 82.5 | 67.5 | 65.8 | 61.2 | 89.8 | 71.6 | 83.4 | 69.3 | 86.2 | 66.7 | 81.7 |
| Deepseek-chat-v2 | 72.7 | 93.1 | 72.8 | **84.9** | **65.7** | **95.1** | **86.7** | **89.2** | **73.5** | 93.8 | **91.9** | **91.1** |
| CoT Prompting | | | | | | | | | | | | |
| Llama-2-7B | 55.0 | 42.6 | 45.7 | 68.9 | 56.5 | 50.8 | 66.7 | 60.1 | 59.5 | 40.5 | 56.7 | 56.1 |
| Llama-2-13B | 66.6 | 55.8 | 63.2 | 40.3 | 59.2 | 60.8 | 71.0 | 56.4 | 60.6 | 61.1 | 63.5 | 74.3 |
| Llama-3-8B | 67.2 | 65.0 | 64.8 | 61.2 | 60.2 | 69.3 | 76.8 | 72.4 | 67.9 | 62.2 | 46.1 | 78.6 |
| Mistral-7B | 61.6 | 64.7 | 65.3 | 58.0 | 58.5 | 64.9 | 73.2 | 77.7 | 60.3 | 66.1 | 69.0 | 79.1 |
| Zephyr-7B-beta | 58.4 | 46.4 | 51.7 | 74.0 | 60.0 | 64.5 | 70.6 | 71.2 | 65.7 | 69.3 | 78.4 | 77.3 |
| Phi-3-mini | 65.1 | 67.9 | 61.9 | 43.0 | 56.5 | 61.2 | 47.8 | 80.5 | 58.4 | 81.2 | 82.6 | 78,7 |
| Phi-3-medium | 69.1 | **82.3** | 72.5 | 58.9 | **63.1** | 89.0 | 87.3 | 79.3 | 54.5 | 81.4 | 87.4 | 85.9 |
| Gemma-2-9B | 73.5 | 74.0 | 71.9 | 81.3 | 62.1 | 89.9 | 85.9 | 82.3 | 63.5 | **83.8** | 87.1 | 69.9 |
| GPT-3.5-0125 | 63.9 | 63.3 | 64.8 | 68.4 | 59.3 | 79.3 | 73.4 | 67.5 | 62.8 | 68.6 | 70.5 | 74.9 |
| Deepseek-chat-v2 | **74.9** | 79.4 | **76.1** | 82.8 | 61.5 | **94.2** | **90.4** | 83.0 | **69.1** | 81.6 | **88.2** | **88.4** |

of consistency does not guarantee similar performance in others. For example, while Mistral-7B excels in transitivity, its performance in other consistency aspects is weaker. The Phi-3 family consistently shows high logical consistency, which we hypothesize is due to its reliance on synthesized data which is cleaner and less self-contradictory.

Another observation is that LLMs tend to perform more consistently on the CaTeRS dataset. We attribute this to the dataset's focus on temporal and causal relations. The more objective and logical preferential relations may make it easier for models to maintain consistency.

**Impact of CoT Prompting.** We also investigated the effect of Chain-of-Thought (CoT) prompting (Wei et al., 2022) on logical preference consistency. Surprisingly, CoT reasoning did not generally improve consistency, and in some cases, it led to a decrease in transitivity performance. We hypothesize that this decline may be due to the additional CoT reasoning tokens could introduce variations in the judgment standards used for different comparisons. When the understanding of the preferential relations is not uniform, it may produce inconsistent (non-transitive) outcomes. This suggests that CoT prompting, while beneficial for complex reasoning, might introduce unintended inconsistencies in certain logical judgment tasks.

### 3.3. Consistency and Reliability

**Transitivity as a Proxy for Self-Agreement.** As shown in Figure 4, there are strong correlations between transitivity and self-agreement across all three datasets, regardless of the task's level of subjectiveness. Self-agreement indicates the model's internal consistency and robustness, as a reliable model should not fluctuate significantly in its responses. Higher self-agreement suggests that the model exhibits a stable understanding of the underlying relations in the input. Given that transitivity captures this self-consistency, we argue that transitivity serves as a useful proxy for evaluating the local robustness of LLMs.

**Commutativity Correlates Strongly with Human Preferences.** For each task, we paraphrased the comparison prompt template 10 times using `GPT-4-turbo` to explore the sensitivity of LLMs to input variations. Despite these paraphrases being semantically equivalent, they produced different performance outcomes, as shown in Figure 5. We observe strong correlations between commutativity and human agreement rates across nearly all datasets and models. This finding is not surprising, as a lack of commutativity often indicates a strong positional bias. These results align with previous research (Zhou et al., 2024), which demonstrated that positional bias can significantly impact align-

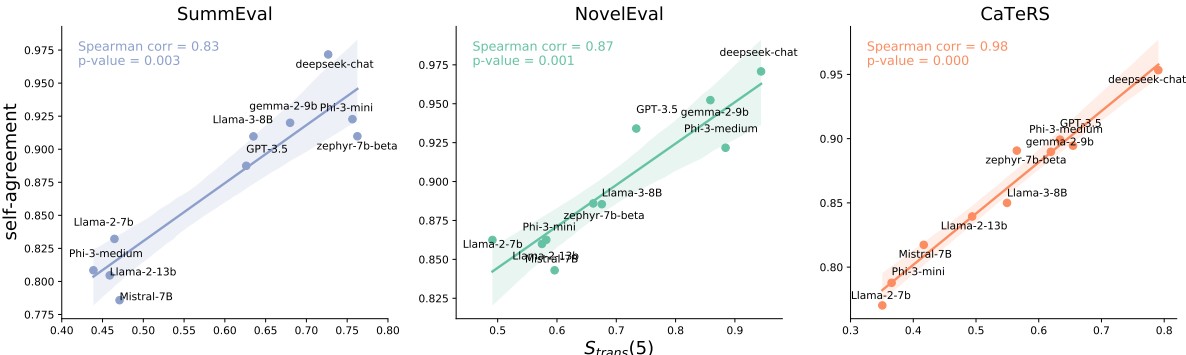

Figure 4: *Transitivity shows strong correlations with self-agreement* across all three datasets. Self-agreement is measured as the percentage of majority choices from 10 CoT inferences, generated with a temperature of 0.7.

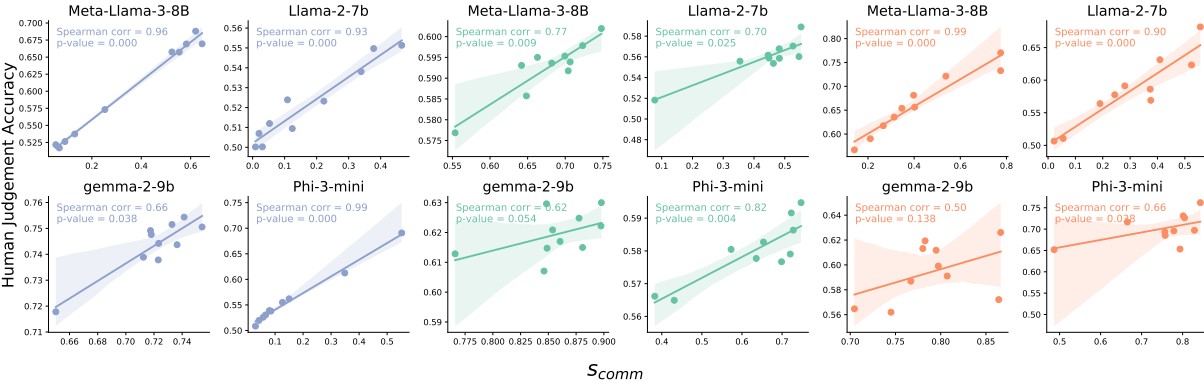

Figure 5: *Commutativity shows a generally strong correlation with human preference* across various LLMs and datasets.

ment with human judgments.

We also show that the strength of this correlation varies between models. For instance, Llama-3-8B exhibits a higher correlation with human preferences compared to Gemma-2-9B. We hypothesize that this difference is due to Gemma-2-9B's training, which was designed to be more robust to input prompts, while other models, like Llama-3-8B, are more fine-tuned to specific prompting styles.

## 4. Improve Logical Preference Consistency in LLMs via REPAIR

Our analysis so far has demonstrated that LLMs exhibit inconsistent logical behavior in making preferential judgments. To address this, we propose REPAIR (**R**anking **E**stimation and **P**reference **A**ugmentation through **I**nformation **R**efinement), a framework that aims to mitigate these inconsistencies. REPAIR first estimates a ranking from noisy preference data, then generates additional conflict-free pairwise comparisons. This approach enhances logical preference coherence while maintaining alignment with human preferences, improving LLMs' reliability as logical operators. See Appendix §K for detailed motivation.

### 4.1. Estimating Rankings from Noisy Pairwise Data

Estimating global rankings from noisy pairwise annotations is essentially a rank aggregation problem. We use the win-loss rate method due to its simplicity and two key advantages: **1)** it performs well with partial and sparse comparisons, which is common in real-world preference datasets, and **2)** it remains unaffected by the order in which comparisons are presented. In Appendix §N, we compare the performance of alternative rank aggregation methods, such as the ELO rating and the Bradley-Terry model, within the REPAIR framework. Additionally, we provide a detailed analysis of their respective strengths and limitations.

The win-loss rate for each item is calculated as the number of comparisons it wins minus the number of comparisons it loses and then divided by the number of comparisons it participates in. Following that, as shown in Figure 6, we rank the item by the value of its win-loss rate. Through this way, a full or partial ranking is aggregated. We then extrapolate the ranking into a self-consistent pairwise comparison set, which can be further augmented by adding comparisons with the negated relation.

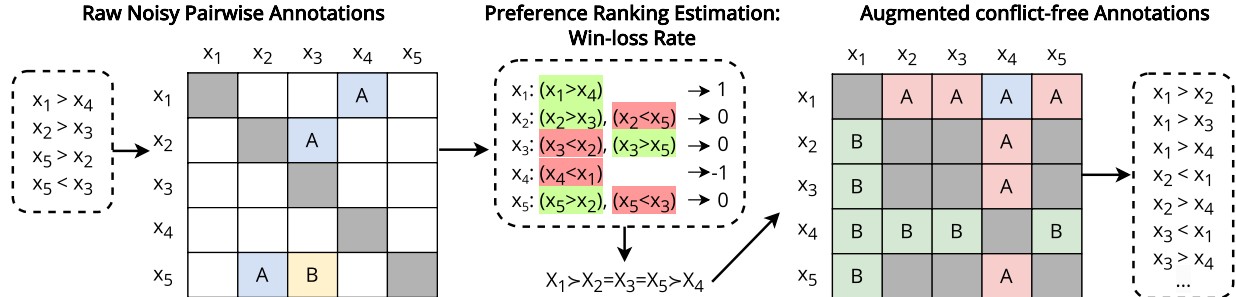

Figure 6: *Illustration of data refinement and augmentation* based on preference rank estimation using win-loss rates. The original annotations (left) are sparse and exhibit inconsistencies (non-transitivity), while the augmented annotations are consistent and include *more* comparisons. For partially ordered rankings, items are not compared when their preference relationship is unknown.

Table 2: Statistics at each augmentation stage. # Data means data size and AvgComp/Inst for average comparisons per instance. REPAIR-ed data represents data augmented by our REPAIR framework and Neg. denotes further augmentation with negated relation comparisons.

| Data | $s_{tran}(5)$ | $s_{comm}$ | # Data | AvgComp/Inst |
|---|---|---|---|---|
| Raw data | 98.4 | - | 14.8K | 6.29 |
| Perturbed data | 87.6 | - | 14.8K | 6.29 |
| REPAIR-ed data | 1 | 1 | 30.9K | 13.2 |
| REPAIR-ed data+Neg. | 1 | 1 | 61.8K | 26.4 |

Table 3: Instruction-tuning of Llama-3 Instruct (8B) using different variants (see Section 4.2). Only training on negated relations can enhance negation invariance.

| Models | Summarize from Feedback | | | |
|---|---|---|---|---|
| | H. | $s_{tran}(5)$ | $s_{comm}$ | $s_{neg}$ |
| Zero-shot inference | 64.3 | 81.1 | 70.2 | 63.8 |
| Raw data | 74.4 | 93.1 | 89.0 | 60.6 |
| Perturbed data | 70.3 | 91.9 | 88.4 | 61.0 |
| REPAIR-ed data | 70.1 | 95.4 | 91.2 | 60.8 |
| REPAIR-ed data + Neg. | 69.7 | 95.2 | 90.5 | 87.9 |

## 4.2. Experiments

**Experimental Setup.** The previously used datasets (SummEval, NovelEval, and CateRS) were primarily designed as evaluation benchmarks and are therefore too limited in size to support model training. To address this limitation, we employ the Summarize from Feedback dataset (Mostafazadeh et al., 2016b), where annotators made pairwise comparisons between two summaries based on qualitative factors. Further details about this dataset can be found in Appendix §B. Additionally, we provide experiments on another dataset in Appendix §L. Given that original annotations are sparse and relatively clean, we introduce synthetic noise by randomly flipping 10% of the training labels. The objective of REPAIR is to both refine and augment this noisy dataset, thereby enabling the extraction of more reliable training signals. To this end, we apply REPAIR to enhance the consistency and increase the volume of pairwise comparisons, as shown in Table 2.

To assess the effectiveness of REPAIR, we instruction-tuned three Meta-Llama-3-8B-Instruct models on: **1)** the original raw and clean training set data, **2)** the flipped/perturbed training set data, **3)** the refined and augmented dataset, denoted as REPAIR-ed data, and **4)** The REPAIR-ed dataset with additional negated relation comparisons. Training hyperparameters are detailed in Appendix § F. For evaluation, we randomly sample 200 instances from the test set and evaluate all models on this subset. The impact of the data augmentation is assessed in terms of logical consistency and model performance.

**Results and Findings.** Results in Table 3 highlight four key findings: **1)** Zero-shot inference shows considerable logical inconsistency. However, training on perturbed data can already substantially improve both human preference alignment and consistency. **2)** Training on the original, clean dataset provides an upper bound in terms of alignment with human judgments, as it contains the complete and uncorrupted signals. Interestingly, this does not necessarily lead to superior logical consistency when compared to models trained on the REPAIR-ed perturbed data. **3)** Models trained with the REPAIR-ed dataset demonstrate significant gains in transitivity and commutativity, while preserving strong alignment with human preferences. **4)** Only training further on negated relations improves the model performance in negation invariance. However, adding negated relations to the broader dataset may introduce distractions and cause a forgetting effect (Luo et al., 2023), resulting in a slight reduction in performance on other logical properties.

Table 4: LLMs with better transitivity perform better with PairS. Improved commutativity indicates less need for algorithm calibration. For both PairS methods, we report the average Spearman correlations over 100 runs.

| Models | SummEval (Coh) | | | | |
|---|---|---|---|---|---|
| | H. | $s_{tran}(5)$ | $s_{comm}$ | PairS | PairS calibrated |
| Mistral-7B | 63.6 | 95.1 | 59.9 | 27.7 | 31.2 +3.5 |
| Phi-3-mini | 65.6 | 92.8 | 46.9 | 33.9 | 38.0 +4.1 |
| GPT-3.5-turbo | 66.3 | 82.5 | 67.5 | 33.5 | 36.3 +2.8 |
| Llama-3-8B | 67.8 | 91.0 | 76.1 | 37.7 | 38.9 +1.2 |
| Phi-3-medium | 68.8 | 96.2 | 71.0 | 38.9 | 41.3 +.2.4 |

Overall, the findings support the effectiveness of REPAIR in improving the logical preference consistency of LLMs.

## 5. Impact of Logical Preference Consistency on Downstream Applications

LLMs are increasingly used as logical operators in high-level algorithms due to their ability to process and understand text semantics. For instance, Qin et al. (2024) use LLMs to enhance document reranking in information retrieval systems. Similarly, Guo et al. (2024) and Yang et al. (2024) utilize LLMs as optimizers in prompt-tuning algorithms, while Liu et al. (2024) and Liusie et al. (2024b) employ LLMs as pairwise comparators to aggregate preference rankings. When LLMs are used as logical operators, maintaining a high level of logical consistency is critical to ensure predictable and efficient decision-making. In this section, we examine how logical preference consistency influences the performance of LLM-based algorithms in such 'logically grounded' tasks.

We illustrate the impact of logical preference consistency through the Pairwise-Preference Search (PairS) algorithm proposed by Liu et al. (2024). PairS is a sorting-based rank aggregation method that uses LLMs as pairwise evaluators (i.e., a particular *'LLM-as-a-judge'* algorithm), comparing items based on specific attributes using a merge-sort approach. Its performance is measured by comparing the LLM-generated rankings with human judgments via Spearman's correlation. Sorting algorithms depend heavily on logical properties. PairS assumes that LLMs used as comparators have near-perfect transitivity and commutativity for optimal ranking results.

To evaluate this, we conduct controlled experiments on the coherence aspect of SummEval, using various LLMs with similar accuracy to human judgments. As shown in Table 4, we follow Liu et al. (2024) by reporting both raw and calibrated results. The calibration, performed as described in Zheng et al. (2023), averages the evaluation probabilities across both possible pairwise orders. While this technique increases inference cost, it reduces bias by balancing positional preferences. Our findings from Table 4 reveal two key

insights: **1)** Although Phi-3-mini has slightly lower accuracy with human judgments than GPT-3.5-turbo, its stronger transitivity leads to better ranking performance with or without calibration. **2)** There is a clear correlation between an LLM's commutativity and the performance gains from calibration in the PairS algorithm. LLMs that are more commutative rely less on calibration to achieve optimal performance, and therefore, require less computations.

## 6. Related Work

Consistency of LLMs has been explored in several contexts before, where previous research has predominantly focused on two domains: *consistency of factual knowledge* and *entailment consistency* across limited statements.

**Consistency of Factual Knowledge.** Previous work has demonstrated that in knowledge-intensive question-answering tasks, concepts like symmetry and transitivity of knowledge snippets are important (Asai & Hajishirzi, 2020). To this end, a benchmark for studying language model consistency with paraphrased relations (Elazar et al., 2021) has been created. Additionally, different studies have examined whether LLMs have a coherent understanding of knowledge graphs related to real-world entities (Jung et al., 2022; Gu et al., 2023; Kassner et al., 2023; Gu et al., 2023). The 'reverse curse' phenomenon highlights that unlearned factual knowledge cannot be deduced by reversing learned knowledge (Allen-Zhu & Li, 2023; Berglund et al., 2024).

**Entailment and Inference Consistency.** In Natural Language Inference (NLI), the consistency of transitivity and symmetry in statement pairs has been explored using predefined inference rules (Li et al., 2019; Jang & Lukasiewicz, 2023). Jang et al. (2022) proposed a test set to exam the consistency of LLMs for NLI tasks. Prediction consistency has been improved through the use of adversarial first-order logic examples (Minervini & Riedel, 2018).

Despite the wealth of knowledge regarding the logical consistency of LLMs within specific domains, most studies have concentrated on first-order relations—logical connections or implications directly linking two or three individual statements. Consequently, there is a notable gap in research

addressing the consistency and reliability of LLMs when applied to more complex decision-making scenarios or the evaluation of multiple items simultaneously. This limitation suggests a pressing need for further investigation into how LLMs can maintain coherence and consistency in broader, more comprehensive contexts, which is essential for their deployment in practical applications.

## 7. Conclusion

In this work, we explored the critical role of logical preference consistency in enhancing the reliability and trustworthiness of LLMs, introducing a framework to quantify three key properties: transitivity, commutativity, and negation invariance, which serve as strong proxies for assessing overall reliability. To address logical inconsistencies, we proposed REPAIR, a data refinement and augmentation framework, demonstrating that models trained on REPAIR-ed data achieve higher logical preference consistency without compromising human alignment. Additionally, integrating logically consistent LLMs into logic-driven algorithms enhanced global performance and computational efficiency, underscoring the practical benefits of logical coherence in downstream tasks. Our findings highlight logical consistency as a complementary factor to human alignment in developing more reliable LLMs, encouraging its recognition as a fundamental aspect of improving trustworthiness and reliability.

## Impact Statement

There are several potential societal consequences of our work. By ensuring logical coherence in LLMs, we hope to contribute to the development of more reliable and trustworthy AI systems. This could have positive implications for various sectors, including healthcare, finance, and education, where decision-making and reasoning tasks are critical. However, it is important to consider the ethical aspects of deploying such systems. Ensuring logical consistency does not automatically guarantee fairness, transparency, or accountability. Therefore, we encourage the broader community to continue exploring these ethical dimensions in parallel with technical advancements. In summary, while our work aims to advance the field of artificial intelligence by improving the logical consistency of LLMs, we recognize the need for ongoing discussions and research to address the ethical and societal implications of deploying AI systems.

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

## A. Transitivity Calculation

In this section, we show two algorithms: (i) construct a preference relation graph by LLM's judgements, and (ii) detect cycle in a relation graph.

---

**Algorithm 1** Construct Preference Relation Graph

---

1: **Objective:** Construct a relation graph reflecting the understanding of LLM upon a set of items.
2: **Inputs:** $\mathbf{X} = \{x_1, x_2, \ldots, x_N\}$: A set of items; $\text{LLM}(\cdot, \cdot|I)$: LLM, for a given instruction prompt $I$, as a preference operator.
3: **Output:** $G$: A relation graph represented as an adjacency list for each vertex.
4: **Initialize:** Initialize the relation graph $G$ such that for each $x$ in $X$, $G[x] \leftarrow \varnothing$, where $G[x]$ represents the pointed vertexes from vertex $x$.
5: **for** $i \leftarrow 1$ **to** $n$ **do**
6:    **for** $j \leftarrow i + 1$ **to** $n$ **do**
7:       $P(x_i \succ x_j) \leftarrow \text{LLM}(x_i, x_j|I)$
8:       **if** $P(x_i \succ x_j) \geq \frac{1}{2}$ **then**
9:          $G[x_i] \leftarrow G[x_i] \cup \{x_j\}$
10:       **else**
11:          $G[x_j] \leftarrow G[x_j] \cup \{x_i\}$
12:       **end if**
13:    **end for**
14: **end for**
15: **Return** $G$

---

---

**Algorithm 2** Check for Cycles in a Directed Graph

---

1: **Objective:** Detect if a directed graph $G$ contains a cycle.
2: **Inputs:** $G$: A directed graph represented as an adjacency list.
3: **Output:** True if a cycle exists, False otherwise.
4: **Initialize:** $visited \leftarrow \varnothing, recStack \leftarrow \varnothing$
5: **for** each $v$ in $G$ **do**
6:    **if** $v \notin visited$ **and** CycleUtil($v$, visited, recStack) **then**
7:       **Return** True
8:    **end if**
9: **end for**
10: **Return** False

1: **Function** CycleUtil($v$, visited, recStack)
2: $visited \leftarrow visited \cup \{v\}$
3: $recStack \leftarrow recStack \cup \{v\}$
4: **for** each $u$ in $G[v]$ **do**
5:    **if** $u \notin visited \wedge$ CycleUtil($u, visited, recStack$) **then**
6:       **Return** True
7:    **else if** $u \in recStack$ **then**
8:       **Return** True
9:    **end if**
10: **end for**
11: $recStack \leftarrow recStack \setminus \{v\}$
12: **Return** False

---

## B. Dataset

**SummEval** (Fabbri et al., 2021) is a summarization meta-evaluation dataset comprising 100 source texts, each paired with 16 summary candidates generated by various language models (LMs). The dataset is annotated based on four criteria: coherence (COH), fluency (FLU), consistency (CON), and relevancy (RE).

**NovelEval** (Sun et al., 2023) is a document reranking test set consisting of 21 novel questions. This set was constructed by compiling questions and passages from four domains published after the release of GPT-4. For each question, 20 candidate passages were retrieved through Google search, and their relevance was manually labeled.

**CaTerS** (Mostafazadeh et al., 2016b) is a dataset focused on temporal event ordering. It contains 1,600 annotated sentences derived from 320 five-sentence short stories sampled from the ROCStories corpus (Mostafazadeh et al., 2016a). These

stories are organized based on causal and temporal relations. We filtered the dataset to include only instances with seven or more events, resulting in 70 instances.

**Summarize from Feedback.** (Mostafazadeh et al., 2016b) consists of 64,832 summary comparisons drawn from the TL;DR (Cachola et al., 2020) and CNN/DM datasets (Nallapati et al., 2016). This dataset is divided into two sections: pairwise comparisons and axis annotations. We concentrate on the pairwise comparison section, which is labeled based on the overall quality of two summaries. For training, we utilize the entire training set, while for testing, we randomly sample 200 instances from the validation set.

**MS MARCO.** (Nguyen et al., 2016) is a widely used large-scale benchmark for training and evaluating retrieval-based question answering systems. Following the methodology of Sun et al. (2023), we leverage the dataset for a document reranking task. Specifically, they randomly sample a subset of queries from the MS MARCO training set, retrieving 10 candidate passages for each query using BM25. Ranking preferences for these passages were distilled using GPT-3.5-turbo. In this work, we decompose the preference rankings into full pairwise comparison annotations. We randomly sample 1K examples from their query subset and corresponding ranking labels. We then split them into a training set of 800 examples and a test set of 200 examples.

## C. Large Language Models in Evaluation

**Open-Sourced LLMs.** The Llama-2-7b-chat-hf and Llama-2-13b-chat-hf models (Touvron et al., 2023) were developed by Meta and are part of the Llama-2 family. Llama-3-8B-Instruct (Dubey et al., 2024) is another model from Meta's Llama series. The Mistral-7B-Instruct-v0.1 (Jiang et al., 2023) model is an instruction-tuned LLM developed by Mistral AI. The zephyr-7b-beta model (Tunstall et al., 2024) is developed by Zephyr AI. The Phi-3-mini-4k-instruct and Phi-3-medium-4k-instruct models (Abdin et al., 2024) are part of the Phi model series developed by Microsoft. The gemma-2-9b-it (Rivière et al., 2024) model is created by the Gemma team.

**API-Based LLMs.** The GPT-3.5-turbo-0125 model is part of OpenAI's GPT-3.5 series, which is renowned for its strong performance in natural language understanding and generation. Deepseek-chat-v2 (DeepSeek-AI, 2024) is another API-based LLM developed by DeepSeek.

## D. Additional Related Work

**Improving Consistency of LLMs.** Asai & Hajishirzi (2020) proposed an augmentation method and a training regularization term to improve the logical consistency of language models for QA tasks. Minervini & Riedel (2018) proposed an adversarial training procedure to improve the robustness of NLI models to adversarial examples. Wang & Henao (2021) focus on improving the paraphrasing consistency during training for the task of low-resource Named Entity Recognition (NER). Li et al. (2019) proposed a framework to regularize the loss function when training a neural model, based on logic rules to improve the logical consistency.

Kumar & Joshi (2022) performed a symmetric consistency analysis on NLI and semantic textual similarity (STS) tasks in a more conservative manner, arguing that a model should generate not only the same predictions but also the same confidence scores if it is truly input-order invariant. They also observed that pretrained language models violated symmetric consistency and the authors then introduced a consistency regularisation term to compensate for the issue.

**Theory of Consistency.** The concept of transitivity in preferences has been extensively explored, revealing its significance in both psychological and logical frameworks. Tversky (1969) highlighted various psychological factors that can lead to intransitive preferences, illustrating how human decision-making often deviates from rational models. Building on this, Arrow (1950) introduced Arrow's Impossibility Theorem, which addresses the inherent challenges of constructing social choice functions that maintain transitivity.

Further contributions to the understanding of transitivity include a comprehensive review by Regenwetter et al. (2011), which examines both theoretical and empirical perspectives on the subject. Their work underscores the critical role transitivity plays in preference structures. In a more formal context, Tarski (1941) examined the role of transitivity within relational calculus, an area that is foundational to many logical systems. This connection demonstrates how transitivity not only influences preferences but also underpins logical reasoning.

Moreover, Hansson (2001) provided insights into the implications of violating transitivity in preferences, showing that such

violations can lead to logical inconsistencies within decision theory. This theme is further explored by Hyde (2011), who proved that apparent violations of transitivity can give rise to logical paradoxes. Collectively, these works emphasize the importance of transitivity in ensuring consistency, both in decision-making and logical deduction.

## E. Distinguishing Logical Transitivity from Social Network Analysis (SNA) Transitivity

Transitivity is a well-studied concept in Social Network Analysis (SNA). However, the definition and purpose for transitivity in our framework diverge significantly from those in SNA. Below, we highlight the key distinctions:

**Motivations.** In SNA, transitivity measures the tendency of nodes in a network to form tightly connected groups. It is based on real-world social patterns, where "friends of friends" are likely to be friends, leading to clustered communities. In contrast, our logical consistency framework employs transitivity to assess self-conflict in relation graphs generated by models or human judgments. Here, transitivity serves as an indicator of logical coherence rather than network clustering.

**Definition.** In graph theory, transitivity quantifies the formation of closed triangles (fully connected triplets), indicating the probability that two neighbors of a node are also connected. In our framework, however, it measures the extent to which relation graphs are acyclic. Specifically, it evaluates the probability that a randomly selected set of nodes forms an acyclic structure, extending beyond simple triplet closures.

**Application and Scenarios.** SNA transitivity is primarily applied to undirected social network graphs, while our logical transitivity operates exclusively on directed relation graphs. SNA transitivity focuses on the closure of triplets, whereas our logical transitivity evaluates logical circulations spanning 3 to K nodes, with K being the size of the sampled sub-graph. Extending SNA measures to higher-order relationships is nontrivial and computationally inefficient.

To illustrate these differences, we provide a detailed analysis of popular SNA transitivity measures:

1. **Clustering Coefficient** (Wasserman, 1994): Global clustering coefficient measures

$$C = \frac{\text{Number of Closed Triplets}}{\text{Number of All Triplets (closed and open)}},$$

   and local clustering coefficient measures

$$C_i = \frac{\text{Number of closed triplets including node } i}{\text{All triplets including node } i}.$$

   While applicable to directed graphs, clustering coefficient still measures the ratio between closed triplets and total triplets instead of *circular relations* and cannot be applied to larger structures.

2. **Triad Census** (Batagelj & Mrvar, 2001): Counts the relative frequency of 16 possible triadic configurations in directed graphs. However, this method is inherently constrained to triads.

3. **E-I** (External-Internal) Index (Krackhardt & Stern, 1988): Measures the ratio of external to internal ties within a given partition, capturing group closure rather than logical acyclicity.

## F. Instruction-Tuning: Training Details

The model was fine-tuned using the following hyperparameters. A learning rate of $5 \times 10^{-5}$ was employed over the course of 2 training epochs. A weight decay of $1 \times 10^{-2}$ was applied to prevent overfitting. For the LoRA (Low-Rank Adaptation) settings, the rank $r$ was set to 16 and the scaling factor $\alpha$ was set to 64. The batch size during training was 4, with a gradient accumulation step of 2 to effectively handle smaller batches. All training was performed on an A100 machine.

## G. Metric Computation details and Prompt Templates

**Metrics Computation details.** For all three datasets, they have the format of each input context is associated with multiple items, and the task is to determine specific relations among them.

For **SummEval**, the input context is a source text, and the task involves determining coherence preferences among multiple summary candidates. In **NovelEval**, the input context is a query, with the task focused on assessing relevance preferences

| Prompt for **F** (x₁, x₂) | Prompt for **F** (x₂, x₁) | Prompt for $\overline{\textbf{F}}$ (x₁, x₂) |
|---|---|---|
| """

Source Text: {{ input }}

Summary candidate A: {{ output_1 }}
Summary candidate B: {{ output_2 }}

Question: Which summary candidate has better coherence? Please only answer with A or B.

Answer:
""" | """

Source Text: {{ input }}

Summary candidate A: {{ **output_2** }}
Summary candidate B: {{ **output_1** }}

Question: Which summary candidate has better coherence? Please only answer with A or B.

Answer:
""" | """

Source Text: {{ input }}

Summary candidate A: {{ output_1 }}
Summary candidate B: {{ output_2 }}

Question: Which summary candidate has **worse** coherence? Please only answer with A or B.

Answer:
""" |

(a) SummEval

| Prompt for **F** (x₁, x₂) | Prompt for **F** (x₂, x₁) | Prompt for $\overline{\textbf{F}}$ (x₁, x₂) |
|---|---|---|
| """
Query: {{ input }}

Document candidate A: {{ output_1 }}
Document candidate B: {{ output_2 }}

Question: Which document candidate is more relevant to the query? Please only answer with A or B.

Answer: """ | """
Query: {{ input }}

Document candidate A: {{ **output_2** }}
Document candidate B: {{ **output_1** }}

Question: Which document candidate is more relevant to the query? Please only answer with A or B.

Answer: """ | """
Query: {{ input }}

Document candidate A: {{ output_1 }}
Document candidate B: {{ output_2 }}

Question: Which document candidate is less relevant to the query? Please only answer with A or B.

Answer: """ |

(b) NovelEval

| Prompt for **F** (x₁, x₂) | Prompt for **F** (x₂, x₁) | Prompt for $\overline{\textbf{F}}$ (x₁, x₂) |
|---|---|---|
| """
Event context:
{{ input }}

Question: Does {{ output_1 }} happen temporally or causally later than {{ output_2 }}?
Please only answer with 'Yes' or 'No'.

Answer:""" | """
Event context:
{{ input }}

Question: Does {{ **output_2** }} happen temporally or causally later than {{ **output_1** }}?
Please only answer with 'Yes' or 'No'.

Answer:""" | """
Event context:
{{ input }}

Question: Does {{ output_1 }} happen temporally or causally **earlier** than {{ output_2 }}?
Please only answer with 'Yes' or 'No'.

Answer:""" |

(c) CaTeRS

Figure 7: Prompt templates for pairwise comparisons. From left to right: Normal comparison, comparison with reversed item order, and comparison with negated relation.

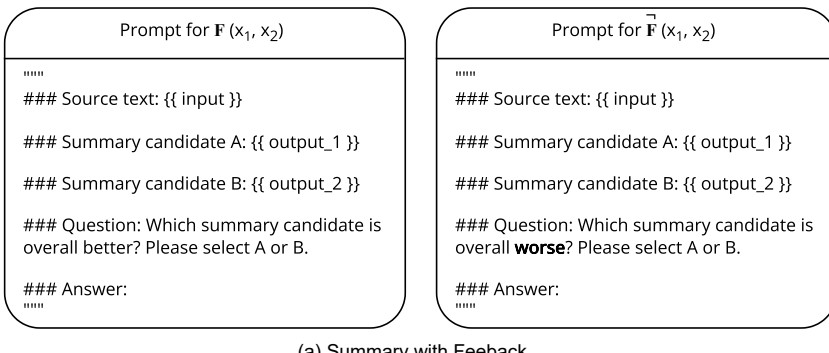

(a) Summary with Feeback

Figure 8: Pairwise comparison prompt templates for the instruction tuning on *Summary with Feedback*, as discussed in Section 4.

among multiple retrieved documents. For **CaTeRS**, the input context consists of a list of unordered events, with the goal being to infer the temporal or causal order among these events.

To compute the consistency metrics, a full pairwise comparison is conducted between every pair of items, including all permutations of the item set $X$. Both standard expression ($F(x_i, x_j)$) and its reverse ($\vec{F}(x_i, x_j)$) are used. This will result two preference matrices as shown in Fig. 3, and can be used to compute the $S_{tran}$, $S_{comm}$ and $S_{neg}$ following the Equ. 1, Equ. 2 and Equ. 3.

**Prompt Templates.** we demonstrate the prompt templates we used to perform pairwise comparisons for all three datasets in Figure 7. Figure 8 illustrates the prompt templates utilized for conducting the instruction tuning experiments on the Summary with Feedback dataset, as detailed in Section 4.

## H. The Choice of K

In this section, we discuss the considerations regarding the choice of $K$ for the transitivity metric $S_{tran}(K)$.

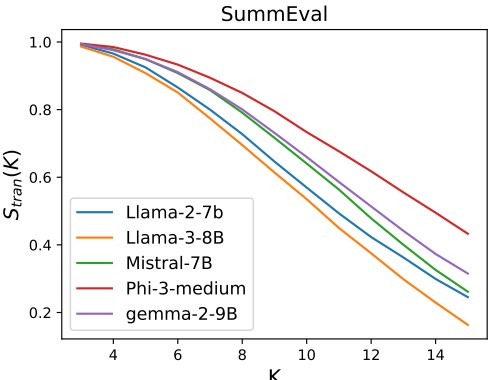

Figure 9: Transitivity metric $S_{tran}(K)$ as a function of sub-graph size, $K$, on the SummEval dataset. Transitivity values decrease as $K$ increases.

**Robustness of transitivity to $K$.** $S_{tran}(K)$ is robust to variations in $K$. It is uncommon for a model to perform well on $S_{tran}(3)$ but poorly on $S_{tran}(5)$, as shown in the Figure 9. Additionally, it shows that larger values of $K$ expand the value range of $S_{tran}(K)$, making comparisons between models more distinct.

**Dependence on Prior Knowledge.** Choosing $K$ can depend on prior knowledge about the general consistency performance of current LLMs. For example, we observe that the consistency performance improves progressively from earlier LLMs, such as LLama-2 and Zephyr-7B-beta, to more advanced models like Gemma-2 and Llama-3. This trend suggests that considering the evaluated LLMs' expected consistency levels can inform an appropriate choice of $K$.

**Task-Specific Considerations.** As discussed in Section 3.2, consistency performance varies depending on the task. For objective tasks, consistency tends to be higher, which means task-specific nuances should also guide the selection of $K$.

In summary, there is no definitive "gold standard" for choosing $K$. Similar to selecting the N-gram size in BLEU-$N$, the choice depends on the models and tasks under evaluation.

## I. Theoretical Evidence for the Choice of transitivity Sub-graph sampling size $M$

**Theoretical Basis.** The transitivity metric $S_{tran}(K)$ is computed as the average result of a binary indicator function (as defined in Equation 1), which follows a binomial distribution. Intuitively, $S_{tran}(K)$ can be interpreted as the probability ($p$) that a randomly sampled $K$-node sub-graph is transitive.

**Estimation Accuracy.** Using the Central Limit Theorem (Barnard, 1949), the binomial distribution of $S_{tran}(K)$ approximates a normal distribution when the sample size ($n$) is sufficiently large. To estimate $p$ within a margin of error ($E$) at a confidence level $z$ (e.g. $z = 1.96$ for 95% confidence), the required sample size is:

$$n = \frac{z^2 \cdot p \cdot (1-p)}{E^2} \tag{4}$$

**Worst-Case Scenario Analysis.** Since the exact value of $p$ is unknown, we consider the worst-case scenario where $p(1-p)$ is maximized at $p = 0.5$. Under this assumption, for a margin of error $E$ at 95% confidence ($z = 1.96$). The sample size $n$ should be at least 394, which is far exceeded by our choice of $M = 1000$. Therefore, our estimation of the $S_{tran}(K)$ is statistically accurate and stable.

## J. The Irreflexive Assumption

As discussed in Section 2, we have stated that if comparison function $F$ is transitive, the corresponding relation graph is a DAG. This statement is based on the assumption that $F$ is irreflexive. For example, in Figure 3 and Equation 2 and 3, we omit the diagonal elements of the pairwise comparison matrices, and we compute the total pairwise comparisons as $|X| \cdot (|X| - 1)$ instead of $|X|^2$.

This assumption is justified by practical considerations: in real-world applications—such as evaluating causal or temporal orders or comparing preferences—it is uncommon to compare an item against itself, particularly in the context of large language models (LLMs).

## K. Motivations for our Data Refinement and Augmentation Framework

Previous work has proven the benefits of using pairwise comparisons derived from preference rankings to train LLMs for a better understanding of preference orderings (Song et al., 2024). Additionally, Asai & Hajishirzi (2020) showed that incorporating logically derived counterparts of pairwise annotations improves consistency in knowledge-intensive QA tasks. However, real-world preference data are often noisy and self-contradictory (Chowdhery et al., 2023; Ouyang et al., 2022), especially in more subjective tasks (Bai et al., 2022). The inter-rater agreement rate typically falls between 60% to 80% for preference or evaluation datasets. Prior work (Wang et al., 2024a) has shown that these self-conflicting annotations reduce the efficiency of learning preferences, and we hypothesize that such annotations also contribute to the logical inconsistencies in trained models. Inspired by RLHF, where a reward model is used to establish a complete order of responses, we propose first estimating a coherent ranking from the noisy data. This refined ranking can then be augmented with additional conflict-free pairwise comparisons to align LLMs more effectively. This method is expected to be more efficient than aligning models to noisy and inconsistent annotations directly.

## L. Performance of REPAIR on the MS-MARCO Dataset

In this section, we present the performance of the REPAIR framework on the MS-MARCO dataset (details provided in Appendix §B). The results, shown in Table 5, correspond to those in Table 3 and Table 4.

Our findings reaffirm the effectiveness of the proposed data augmentation framework. Specifically, we observe that REPAIR enhances consistency while maintaining alignment with human preferences. Additionally, we analyze the Spearman

Table 5: Instruction-tuning of Llama-3-Instruct-8B on the REPAIR-ed *MS MARCO* dataset and their performance on the PairS algorithm. LLMs trained on the REPAIR-ed dataset show improved performance on PairS with reduced need for algorithm calibration. For both PairS methods, we report the average Spearman correlations over 100 runs.

| Models | MS MARCO | | | | | |
| --- | --- | --- | --- | --- | --- | --- |
| | H. | $s_{tran}(5)$ | $s_{comm}$ | $s_{neg}$ | PairS | PairS calibrated |
| Zero-shot inference | 57.2 | 74.2 | 68.5 | 64.7 | 18.7 | 22.1(+3.4) |
| Perturbed data | 74.7 | 86.7 | 81.1 | 62.9 | 58.1 | 60.3(+2.2) |
| REPAIR-ed data | 75.0 | 91.3 | 85.9 | 63.1 | 61.0 | 62.6(+1.6) |
| REPAIR-ed data + Neg. | 74.9 | 90.2 | 86.2 | 85.5 | - | - |

correlations of the rankings predicted by the PairS algorithm. Our results indicate that models with higher transitivity produce more accurate overall rankings, while models with stronger commutativity require less calibration to achieve optimal performance.

## M. Ablation Study: REPAIR Down-Sampling Performance

In this section, we analyze the performance of the REPAIR framework when sampling an equivalent size of perturbed data from the REPAIR-ed dataset. This ensures that the comparison remains unaffected by dataset size differences, allowing for a clearer assessment of the framework's efficacy.

REPAIR is a data augmentation framework, a widely adopted strategy to mitigate the challenges of acquiring high-quality, consistent data, particularly in the era of LLMs. Given the increasing difficulty and resource intensity associated with obtaining such data through synthetic methods or human annotations, data augmentation offers a practical and cost-effective alternative. Our framework enhances data quality solely by leveraging the perturbed dataset, without requiring any external knowledge.

Although the datasets differ in size, the comparisons in Table 3 remain valid, as they effectively showcase the ability of our approach to extract and expand information from the perturbed data.

To conduct this ablation study, we randomly sample an equivalent amount of REPAIR-ed data as the perturbed dataset while maintaining all other experimental settings consistent with those in Table 3. The experimental results, summarized in Table 6, yield the following key observations:

1. **Information Expansion and Loss:**
   - The REPAIR-ed dataset enriches the information content of the perturbed dataset. However, random sampling from this augmented dataset inevitably leads to some degree of information loss.
   - Without employing a targeted sampling strategy to mitigate this loss, the comparison may not fully reflect the optimal benefits of our augmentation approach.

2. **Impact on Performance:**
   - Due to the inherent information loss from random sampling, alignment with human preferences exhibits a slight decline. However, this performance drop remains relatively minor.
   - Despite the reduced dataset size, all three consistency metrics demonstrate noticeable improvements over the perturbed dataset. This highlights the robustness of our augmentation framework, even though the performance is naturally lower than that of models trained on the full augmented dataset.

These findings reaffirm the efficacy of our proposed framework and offer deeper insights into its behavior under constrained data conditions.

## N. Ablation Study: Alternative Ranking Estimation Methods for REPAIR

In the main paper, we presented REPAIR's performance using the win-loss rate as the ranking estimation method. In this section, we conduct additional experiments to compare win-loss rate with alternative ranking methods.

Table 6: Instruction-tuning of Llama-3-Instruct-8B on the down-sampled REPAIR-ed *Summarize from Feedback* dataset, matched in size to the Perturbed data. Despite information loss from the down-sampling, all three consistency metrics show steady improvements over the perturbed dataset.

| Models | Summarize from Feedback | | | |
|---|---|---|---|---|
| | H. | $s_{tran}(5)$ | $s_{comm}$ | $s_{neg}$ |
| Zero-shot inference | 62.9 | 84.7 | 68.2 | 64.0 |
| Perturbed data | 71.3 | 92.4 | 79.6 | 60.9 |
| REPAIR-ed data (sampled) | 69.1 | 96.3 | 88.9 | 61.7 |
| REPAIR-ed data + Neg. (sampled) | 68.8 | 96.2 | 86.2 | 84.5 |

Table 7: Instruction-tuning of Llama-3-Instruct-8B on the REPAIR-ed *Summarize from Feedback* dataset using various rank estimation methods. All methods integrate well with the REPAIR framework and yield notable improvements.

| Models | Summarize from Feedback | | |
|---|---|---|---|
| | H. | $s_{tran}(5)$ | $s_{comm}$ |
| Zero-shot inference | 62.9 | 84.7 | 68.2 |
| Perturbed data | 71.3 | 92.4 | 79.6 |
| REPAIR-ed data **W-L** | 70.5 | 97.5 | 89.4 |
| REPAIR-ed data **ELO** | 71.4 | 97.7 | 89.7 |
| REPAIR-ed data **B-T** | 71.2 | 98.6 | 91.0 |

We evaluate two widely used ranking methods: the Elo rating system (Elo & Sloan, 1978) and the Bradley-Terry (B-T) model (Bradley & Terry, 1952). Table 7 reports the results from these methods, maintaining consistency with the experimental setup in Table 3 (Section 4). Our findings indicate that both Elo and the B-T model achieve slightly better performance than the win-loss rate, improving human accuracy and consistency metrics. However, the performance differences remain relatively small.

To further analyze the strengths and weaknesses of ranking estimation methods, we summarize their characteristics below:

- **Win-loss rate:** This method does not incorporate transitive information, treating all pairwise comparisons independently. While this may appear to be a limitation, it has certain advantages:
  - **Pros:** Consider the pairwise comparisons $\{A \succ B, A \succ C\}$. The win-loss rate does not infer a relationship between B and C, correctly predicting $A \succ B = C$. Later, we show how Elo and B-T fail in such cases.
  - **Cons:** In a transitive scenario $\{A \succ B, B \succ C, C \succ D\}$, an ideal ranking should yield $A \succ B \succ C \succ D$. However, the win-loss rate produces $A \succ B = C \succ D$, as it does not consider indirect comparisons.

- **Elo rating:** This method effectively incorporates transitive information but is highly sensitive to the order of comparisons. When pairwise comparisons are sparse, this sensitivity can significantly impact ranking estimation. For example:
  - Given $\{A \succ B, A \succ C\}$, Elo predicts $A \succ C \succ B$.
  - If the order is reversed to $\{A \succ C, A \succ B\}$, Elo instead predicts $A \succ B \succ C$.

  Since our dataset is static and comparison order carries no inherent meaning, Elo's sensitivity may introduce noise into the augmented preference data.

- **Bradley-Terry model:** This probabilistic model estimates rankings using maximum likelihood, effectively handling transitive relationships and generating nuanced rankings. However, it has limitations:
  - It struggles to produce tied rankings without explicit design modifications. For instance, given $\{A \succ B, A \succ C\}$, the B-T model predicts $A \succ B \sim C$, where B and C receive slightly different scores, even if they should theoretically tie.
  - This forced ranking structure may introduce noise in scenarios where ties should be preserved.

Overall, the effectiveness of each ranking estimation method depends on factors such as pairwise comparison sparsity and computational constraints:

- **Sparse comparisons:** Methods relying on transitive relationships, such as Elo and B-T, may struggle when comparison data is limited.

- **Computational efficiency:** Simpler methods like the win-loss rate are advantageous when computational resources or time are constrained.

These observations align with the axiomatic assumptions of the Von Neumann–Morgenstern utility theorem (**?**), which requires that preferences be complete—that is, a preference judgment must exist between every pair of items—to aggregate preferences into a consistent linear ranking. When comparison annotations are sparse, this completeness assumption is violated, leading to a degradation in the performance of ranking estimation methods.

In summary, while Elo and the B-T model are effective in capturing transitive relationships and generating likelihood-based rankings, their sensitivity to noise and computational demands can be limiting in certain contexts. In contrast, the win-loss rate offers a simple and robust baseline, especially when tied rankings are acceptable or when efficiency is a key concern. Additionally, extensions such as TrueSkill build upon Elo and B-T by explicitly modeling ties and uncertainty, potentially mitigating some of these limitations.

