# OpenReview forum: "Aligning with Logic: Measuring, Evaluating and Improving Logical Preference Consistency in Large Language Models"
_ICML.cc/2025/Conference — ICML 2025 spotlightposter_

### Official Review · Reviewer_paNc · 2025-03-11

**Overall Recommendation:** 3

**Summary:**

The paper introduces a novel framework for enhancing the logical consistency of large language models (LLMs). The authors propose a universal evaluation framework based on three key properties: transitivity, commutativity, and negation invariance. They also introduce REPAIR, a data refinement and augmentation method that improves logical consistency while maintaining alignment with human preferences. The study demonstrates significant improvements in model performance across various tasks, including abstractive summarization evaluation, document reranking, and temporal event ordering. The main findings include substantial enhancements in logical consistency metrics and better performance in logic-dependent algorithms.

**Claims And Evidence:**

The claims made in the submission are well-supported by clear and convincing evidence. The authors provide extensive experimental results across multiple datasets (SummEval, NovelEval, CaTeRS) to demonstrate the effectiveness of their proposed framework. The metrics used (transitivity, commutativity, negation invariance) are clearly defined and appropriately applied to evaluate logical consistency. The results show significant improvements in logical consistency and performance, which validate the claims made in the paper.

**Essential References Not Discussed:**

To my knowledge, I do not find any other related work essential to understanding the key contributions of the paper, but not currently cited/discussed in the paper.

**Experimental Designs Or Analyses:**

The experimental designs and analyses are valid. The authors conduct extensive experiments across multiple datasets to evaluate the effectiveness of REPAIR. The results are presented in a clear and structured manner, with detailed comparisons to baseline methods. The experimental setup is appropriate for the tasks considered, and the results support the claims made in the paper.

**Methods And Evaluation Criteria:**

The proposed methods and evaluation criteria are well-suited for the problem at hand. The authors use a combination of ranking estimation and data augmentation (REPAIR) to enhance logical consistency. The evaluation criteria, including transitivity, commutativity, and negation invariance, are relevant and effectively measure the logical coherence of LLMs. The use of benchmark datasets (SummEval, NovelEval, CaTeRS) makes sense for evaluating the proposed methods.

**Other Comments Or Suggestions:**

The paper could benefit from additional visualizations or examples to illustrate the effectiveness of the REPAIR method.

The authors may ensure that the code and refined datasets are publicly available to facilitate reproducibility.

**Other Strengths And Weaknesses:**

Strengths:

The paper presents a novel and comprehensive framework for evaluating and improving logical consistency in LLMs.

The proposed REPAIR method is innovative and shows significant improvements in logical consistency and performance.
The experimental results are robust and validate the effectiveness of the proposed methods.

Weaknesses:

The paper could be strengthened by exploring more potential limitations of the proposed methods, such as their applicability to other types of models or tasks.

**Questions For Authors:**

Limitations and Future Work: What are the potential limitations of the proposed methods, and what are the key areas for future research? How do the authors plan to address these limitations?

Comparison to Adversarial Training: How does REPAIR compare to adversarial training methods (e.g., Minervini & Riedel, 2018) in terms of improving logical consistency? Are there any specific advantages or disadvantages?

**Relation To Broader Scientific Literature:**

The key contributions of the paper are well-aligned with the broader scientific literature on improving logical consistency in neural models. The paper builds on prior work in natural language processing (NLP) and machine learning (ML) that focuses on enhancing model reliability and coherence. Specifically, the paper relates to works such as:

Adversarially Regularising Neural NLI Models to Integrate Logical Background Knowledge (Minervini & Riedel, 2018): This work investigates the problem of generating adversarial examples to improve logical consistency in NLI models.

Logic-Guided Data Augmentation and Regularization for Consistent Question Answering (Asai & Hajishirzi, 2020): This study integrates logical rules with neural models to improve consistency in QA tasks.

**Theoretical Claims:**

The paper does not present any formal theoretical proofs. However, the conceptual framework and the proposed methods are logically sound. The authors provide a clear rationale for the choice of logical consistency properties (transitivity, commutativity, negation invariance) and their application to LLMs. No specific theoretical claims require further validation.

---

> ### Author Rebuttal · Authors · 2025-04-01
>
> We sincerely appreciate the reviewer's thoughtful feedback and acknowledgment. Below, we provide our clarifications and explanations.
>
> ---
>
> - Potential Limitations of the proposed methods:
>
>   In the appendix, we analyze the limitations of different rank estimation methods. Another potential limitation is that the proposed REPAIR method may not be effectively applied to preference data when there are only two responses per query (i.e., when the data is too sparse). In such cases, the rank estimation method cannot reliably distinguish between true preference and noisy preference.
>
> ---
>
> - Future direction:
>
>     A promising future direction is to investigate whether the improved logical consistency achieved by REPAIR can be generalized. For example, if we augment a general-purpose query-response preference dataset using REPAIR, would training on this augmented dataset improve logical consistency in other domains? To what extent? While we believe these are valuable research directions, they require more exhaustive investigations, which are beyond the scope of this work.
>
> ---
>
> - Additional visualizations:
>
>   We will make every effort to improve the clarity of the REPAIR-related figures to enhance their readability and comprehension.
>
> ---
>
> - Code and dataset release:
>
>     We confirm that we will release all code and datasets to facilitate reproducibility and further research.
>
> ---
>
> - Comparison to adversarial training (Minervini & Riedel, 2018)
>
>   We appreciate the reviewer for pointing out this relevant work. The primary differences between Minervini & Riedel (2018) and our REPAIR method are as follows:
>
>   - The previous work focuses solely on the Natural Language Inference (NLI) task, where a well-defined logical relationship of entailment exists. In contrast, our framework is more general and applicable to any rankable items, including entailment relations, causal relations, temporal orders, and preference orders.
>
>   - The prior work mitigates inconsistency by adding regularization to the loss function, whereas REPAIR focuses on a data augmentation approach to address inconsistency. As discussed in the Related Work section and Appendix D, other data-centric methods exist, but they are often restricted to NLI relations or factual knowledge consistency, which limits their applicability to broader domains.
>
> ---
>
> We will revise the manuscript according to the reviewer's suggestions and hope we have addressed all the concerns.

---

### Official Review · Reviewer_hAwv · 2025-03-13

**Overall Recommendation:** 1

**Summary:**

This paper investigates whether large language models (LLMs) make pairwise preference judgments without logical contradictions, focusing on three consistency properties: transitivity (A > B > C implies A > C), commutativity (choices remain the same regardless of order/phrasing), and negation invariance (consistent answers when preferences are reversed or negated). The authors propose metrics for each property and show that state-of-the-art LLMs often violate them, despite aligning well with human judgments. To address this, they introduce REPAIR, which aggregates noisy preference data into a self-consistent global ranking and then adds logically implied or negated comparisons for fine-tuning. REPAIR raises consistency scores without jeopardising human alignment and boosts performance in downstream LLM-as-judge tasks.

**Claims And Evidence:**

The paper makes several claims supported by empirical evidence to certain extent:
- Logical preference consistency (as defined by transitivity, commutativity, and negation invariance) is crucial for reliable LLM decision-making, yet current models often exhibit inconsistencies.
- The proposed consistency metrics serve as indicators of model robustness and alignment, showing strong correlations with human preference agreement rates.
- The REPAIR method effectively improves an LLM’s logical consistency without compromising alignment with human preferences.
- Models with higher logical consistency perform better in downstream tasks that require iterative judgments, such as ranking algorithms.

However, the paper overstates its contributions by describing the evaluation framework as "universal". In reality, the study is specific to pairwise preference-ranking scenarios, and its applicability to broader LLM tasks remains unverified. Also, the experiments do not sufficiently support the claims (see one critical aspect in Experimental Designs Or Analyses).

**Essential References Not Discussed:**

This submission largely builds upon [1], which is cited. However, it does not sufficiently discuss how it improves over that prior work, leaving unclear whether its contributions are meaningful advancements or merely incremental refinements.

[1] Liu, et al. "Aligning with human judgement: The role of pairwise preference in large language model evaluators." COLM (2024).

**Experimental Designs Or Analyses:**

An critical issue lies in the experiment design is that the authors conduct their logical consistency evaluation on three datasets (SummEval, NovelEval, CaTeRS) but then demonstrate the effectiveness of their REPAIR method on a different dataset (Summarize From Feedback). This raises several concerns and questions:

- Did the fine-tuned model show improved logical consistency when re-evaluated on SummEval, NovelEval, and CaTeRS?
- If so, why were those results not reported?
- If not, does this suggest that REPAIR’s improvements are dataset-specific or do not generalise across tasks?

**Methods And Evaluation Criteria:**

The methodology introduced in this paper is mostly appropriate for the stated problem. The authors identify transitivity, commutativity, and negation invariance as key measurable aspects of logical consistency in pairwise judgments, and they formalise metrics for each. These choices are sensible: transitivity directly addresses the avoidance of preference cycles, commutativity tests that the order or wording of input does not flip the outcome, and negation invariance ensures a model handles logically equivalent inquiries consistently.

However, one particular thing remains arguable is the use of win-loss rate in its REPAIR algorithm:
- Win-loss treats all preferences equally, whereas in real-world data, some preferences may be stronger or more certain than others. Methods that incorporate confidence scores or margins of preference may produce more reliable rankings.
- If a model produces inconsistent rankings (e.g., due to randomness or bias), win-loss does not provide an effective way to smooth or account for contradictions.
- Win-loss does not estimate a likelihood of preference correctness but rather a raw win ratio, which lacks formal statistical grounding.

**Other Comments Or Suggestions:**

- There appears to be a contradiction in Figure 6’s right-most subfigure, where the preference matrix entry $(X_2, X_4)$ is labeled "A" (indicating $X_2 > X_4$), but the side prompt states $X_2 < X_4$ (indicating the opposite relationship).

**Other Strengths And Weaknesses:**

n/a

**Questions For Authors:**

My main questions and concerns have been discussed above.

**Relation To Broader Scientific Literature:**

As illustarted, this is largely within the scope of preference ranking in LLMs.

**Theoretical Claims:**

This work is largely empirical. One nuanced theoretical claim is its choice of sampling sub-graphs for transitivity check.

---

> ### Author Rebuttal · Authors · 2025-04-01
>
> We sincerely appreciate the reviewer's thoughtful feedback. Below, we provide our clarifications and explanations.
>
> ---
>
> - Overstatment on "universal framework":
>
>   We agree that our work focuses on the logical preference consistency of pairwise preference-ranking scenarios. However, we would like to clarify that **by "universal framework" we mean that our proposed method applies across all domains with multiple rankable items**, unlike previous works that primarily focus on tasks such as NLI and factual knowledge consistency.
>   If the reviewer still finds "universal framework" to be an overstatement, we are happy to revise or remove the term accordingly.
>
> ---
>
> - Concerns about the Win-loss rate rank estimation method
>
>   We acknowledge the potential limitations of using the win-loss rate as a rank estimation method. However, we would like to refer to our **ablation study in Appendix N** (referenced in the main paper, Line 326), where we provide **detailed analysis of the advantages and disadvantages of two alternative rank estimation methods, ELO rating and Bradley-Terry model**. Additionally, we compare REPAIR's performance across different rank estimation methods. We believe this ablation study sufficiently addresses the concerns raised. However, if further clarification is needed, we would be happy to expand on this discussion.
>
> ---
>
> - Concerns about experimental design
>
>   The use of different datasets for the REPAIR method (Summary from Feedback and MS MARCO, Appendix L) and for the quantification method (SummEval, NovelEval, and CaTeRS) is intentional. The latter datasets are designed as evaluation benchmarks and are too small to be split for training.
>
>   For details on dataset sizes, please refer to Appendix B.
>
> ---
>
> - Concerns about Generalization:
>
>   We believe the reviewer is questioning whether the learned consistency can generalize across different domains. We would like to clarify that **we do not claim that the REPAIR method improves consistency across domains**. While we agree that cross-domain generalization is an important future direction, it is beyond the scope of this paper.
>
>   Systematically justifying such a claim is challenging due to the multiple factors influencing generalization performance, such as the specific preference learning method (e.g., DPO or PPO) and the domain gap between training and target tasks. For example, training on summaries with feedback may improve SummEval but not necessarily document reranking tasks.
>
>   If the reviewer is concerned about potential overstatement, we are open to revising our claim to specify "improving in-domain logical consistency."
>
> ---
>
> - Comparison to previous work PairS[1].
>
>   We would like to clarify that **our work and PairS address completely different research problems and are not directly comparable**. PairS focuses on ranking estimation via search-based methods, whereas our paper aims to **quantify and improve the logical preference consistency of LLMs**.
>
>   The only connection to PairS appears in Section 5, where we use it to illustrate **how logical preference consistency impacts real-world logic-dependent algorithms**.
>
> [1] Liu, et al. "Aligning with human judgement: The role of pairwise preference in large language model evaluators." COLM (2024).
>
>
> ---
>
> - Error in Figure 6
>
>   We thank the reviewer for spotting this typo. We will revising it accordingly.
>
> ---
>
> We appreciate the reviewer's feedback and hope our clarifications address their concerns. We are open to revising our claims and further refining our discussion if needed.

---

### Official Review · Reviewer_3jpj · 2025-03-13

**Overall Recommendation:** 3

**Summary:**

This paper introduces a universal framework for evaluating logical preference consistency in LLMs, focusing on three properties: transitivity, commutativity, and negation invariance. The authors propose quantitative metrics for these properties, conduct comprehensive empirical analyses across state-of-the-art LLMs, and introduce REPAIR, a method for refining and augmenting data to improve logical consistency. Experimental results indicate enhanced consistency correlates with better model reliability and improved performance in logic-dependent tasks.

**Claims And Evidence:**

Yes

**Essential References Not Discussed:**

No

**Experimental Designs Or Analyses:**

Yes

**Methods And Evaluation Criteria:**

Yes

**Other Comments Or Suggestions:**

1. The paper lacks a detailed ablation study on components of the REPAIR method.
2. Clarify more explicitly how human judgments were utilized in the data augmentation step.
3. It would be useful to clarify the computational overhead introduced by the REPAIR method.

**Other Strengths And Weaknesses:**

Strengths:
1. Clearly defined formulations and rigorous metrics for evaluating logical consistency.
2. Extensive experimentation across diverse LLMs and multiple tasks validates the utility of proposed metrics and methods.
3. REPAIR method effectively improves consistency without sacrificing alignment with human preferences. Solid empirical analysis showing strong correlations between proposed consistency metrics and model reliability.

Weaknesses:
1. Evaluation is limited to datasets with relatively well-structured logical tasks; extending evaluation to more complex or noisy real-world applications could strengthen claims of universality.
2. The observed negative effects of Chain-of-Thought prompting on logical consistency are intriguing but insufficiently explored; deeper analysis or theoretical insight would be valuable.

**Questions For Authors:**

1. Have you explored the effectiveness of the proposed method on larger-scale and less structured datasets (e.g., open-ended reasoning tasks)?
2. How sensitive is your consistency metric to variations in annotation quality?
3. Could you discuss potential trade-offs in consistency versus expressivity or creativity of the model outputs?

**Relation To Broader Scientific Literature:**

The paper improves reliability of LLMs by addressing logical consistency.

**Theoretical Claims:**

N/A

---

> ### Author Rebuttal · Authors · 2025-04-01
>
> We sincerely appreciate the reviewer's thoughtful feedback and acknowledgment. Below, we provide our clarifications and explanations.
>
> - Extending the Experiment and Method to Less Structured Datasets
>
>   We focus on analyzing logical preference consistency, and by '**universal** framework' we mean that our proposed method applies across **all domains with multiple rankable items**. As stated in Line 218, our dataset selection is guided by **subjectiveness** considerations: SummEval (evaluating summaries) represents a more subjective task, whereas CaTeRS (comparing temporal and causal order) is a more objective one. We believe covering this spectrum of subjectivity ensures broad applicability to real-world tasks. However, if the term "universal framework" seems overstated, we are open to refining it to "improve in-domain consistency."
>
> ---
>
> - CoT Analysis
>
>   Thank you for raising this point. Our findings show that evaluation with **CoT reasoning enhances accuracy but does not necessarily improve consistency**. While we recognize that systematically analyzing CoT’s impact on LLM judgments is a valuable research direction, conducting an exhaustive set of experiments is beyond the scope of this paper. Our primary focus is on **quantifying consistency, assessing its impact, and proposing methods for improvement**.
>
> ---
>
> - REPAIR Method: Ablation Study on Components
>
>   We appreciate the reviewer's suggestion. The REPAIR method consists of two key steps:
>   1) **Rank estimation** from noisy and sparse human annotations.
>   2) **Preference augmentation** based on the estimated rankings.
>
>   In Appendix M and N, we conduct two ablation studies:
>   - **Evaluating different rank estimation methods.**
>   - **Assessing the performance impact of using the same amount of augmented data.**
>
>   To the best of our knowledge, these cover all potential ablation studies relevant to the REPAIR method.
>
> ---
>
> - REPAIR Method: Use of Human Judgments in Data Augmentation and Computational Overhead.
>   As illustrated in Figure 6, **human judgment annotations are solely used in the first step of REPAIR, where a ranking is estimated from the noisy human annotations**. We will improve the visualization and explanation to make this clearer. Regarding computational overhead, since REPAIR is a data augmentation method, its **additional computational cost is directly proportional to the size of the augmented dataset**. Table 2 (# data, Avg Comp/Inst) details the amount of data generated by REPAIR and its computational implications.
>
> ---
>
> - Sensitivity of Consistency Metric to Annotation Quality
>
>   Our consistency metrics—transitivity, commutativity, and negation invariance—**do not rely on human annotations**. Instead, they are derived solely from the model’s own judgments, making them independent of annotation quality.
>
> ---
>
> - Trade-offs Between Consistency and Creativity in Model Outputs.
>
>   This is an excellent question. We believe that consistency and creativity are distinct dimensions of LLM behavior, each desirable in different applications. For instance, when simulating human behavior, perfect logical consistency may not be necessary for all queries. Therefore, **we do not advocate for universal consistency improvements across all domains. Instead, we propose that when logical consistency is a critical factor for an LLM’s intended functionality**, e.g. used as an logical operator in a logic-dependent high-level algorithm, our method provides a systematic way to quantify and enhance this characteristic.
>
> ---

---

### Official Review · Reviewer_sp8E · 2025-03-14

**Overall Recommendation:** 4

**Summary:**

The main topics of this paper are (1) introducing consistency metrics for several logic/preference orders of LLMs, (2) evaluation of a large number of models and datasets, and (3) impact on downstream applications.
Specifically, the metrics measure transitivity, commutativity, and negation invariance in LLMs based on in-context learning, as shown in Figures in the Appendix, such as Figure 7. The proposed data augmentation scheme, REPAIR, also enhances the consistency behavior.

**Post rebuttal**
I checked the paper again and read reviews from other reviewers.
One reviewer had a different evaluation (1) than others (4/3/3).
In short, two reasons for rejection were (1) related to the citation of one paper and (2) essentially, asking for more experiments.
I wanted to clarify how serious these issues are, but I couldn't find substantial evidence supporting such arguments.
Therefore, I will maintain my original score.

**Claims And Evidence:**

The main claim is that given LLM preference orders under some relations, the proposed method measures the transitivity, commutativity, and negation consistency over the pairs of the statement.  The formulation is convincing.

The evaluation of LLMs shows the scores from various models over different datasets.
The outcome in Table 1 is also convincing.

Some interesting hypotheses in Section 3.3, such as transitivity as a proxy for self-agreement, commutativity correlates with human preferences, are also provided with the evidence in Figure 4 and 5.

**Essential References Not Discussed:**

NA

**Experimental Designs Or Analyses:**

Overall, the experiment is well designed, and the appendix supports its validity and soundness.

**Methods And Evaluation Criteria:**

Yes, the definition of metrics, selection of the models, and datasets make sense.

**Other Comments Or Suggestions:**

NA

**Other Strengths And Weaknesses:**

The main strength is the empirical work that supports various claims and reproducible details in the paper.

**Questions For Authors:**

I don't have any further questions.

**Relation To Broader Scientific Literature:**

This empirical result might impact the consistency behavior around LLMs. The observation made in this paper is consistent with the earlier findings in smaller LMs and matches a new trend over very large LMs.

**Theoretical Claims:**

I believe there is no theoretical claim in this paper.

---

> ### Author Rebuttal · Authors · 2025-04-01
>
> We sincerely appreciate the reviewer's feedback and acknowledgment. We are grateful for the time and effort dedicated to reviewing our work.
>
> Best,
> Authors

---

### Decision · Program_Chairs · 2025-05-01

**Decision:**

Accept (spotlight poster)

**Comment:**

The paper provides an important, interesting, and timely evaluation of preference consistency for LLMs given the prevalence of LLM-based WinRate comparison-based evaluation in a very broad range of experimental settings.  The paper further goes on to provide a repair method to improve preference consistency.  Appendices provide many additional interesting evaluations and results to augment the primary contributions of the paper.

Three out of four reviewers vote for acceptance, while one reviewer raises experimental concerns and recommends rejection.  I understand and agree with the concerns of that reviewer (as discussed further below), but having read the paper, I don't think this experimental issue alone is sufficient to reject the paper in light of it's novelty, insights, and otherwise expansive and diverse experimental evaluation.

The authors are strongly encouraged to take reviewer feedback into account to improve clarity and explanation when revising the paper.  I highlight one particular point of concern I share with a reviewer along with three additional points I would like to raise that can hopefully improve and enhance the paper presentation:

1. One reviewer and I retain a concern that there appears to be an unjustified shift (in the present paper) between the datasets used in Section 3 and Section 4.  The authors provide a response regarding this in their rebuttal.  The reviewer who initially raised this issue is not convinced, nonetheless I suggest that the authors include their justification on revision so they can make their experimental rationale clear.

   To further elaborate on concerns about this experimental shift, I don't understand why three datasets and many LLMs are presented initially but the proposed method is only applied to one dataset ("Summarize from Feedback") with only one LLM, and even then (line 365) they sample 200 test instances randomly, which they do not report doing for Table 1.

2. In Table 3 the authors test a (single) Llama-3 Model fine tuned on the "Summarize from Feedback" dataset (it is not clear if they are using a dedicated train set, which should be clarified) which has been augmented in three ways: (1) perturbed by corrupting 10% of labels, (2) REPAIRed, and (3) REPAIRED with negative examples, but they do not report the results of a tuning the Llama model on the original training data, which is an important comparison and unexplained omission. The comparison between their methods (2), (3), a zero-shot model and (1) which is a corrupted dataset does not seem like a fair or informative comparison.  More explanation is needed to justify this experiment and its conclusions.

3. Seemingly out of the blue, results on MS-Marco reranking appear in the Appendix in what appears to be a reranking task, but it is not clear to me where the initial list being reranked comes from (BM25?). The authors somehow apply their REPAIR method to augment MS-Marco labels, but labels in MS MARCO are not pairwise comparisons or preference orderings on passages/documents, they are pointwise relevance judgements, so it is extremely unclear what the authors are doing here.  This requires clarification.

4. I remark that when converting preferences to a score-based total order ranking (e.g., ELO, Bradley-Terry) seems to implicitly require the [Von Neumann–Morgenstern utility theorem](https://en.wikipedia.org/wiki/Von_Neumann%E2%80%93Morgenstern_utility_theorem) to verify that such a score-based utility exists along with it's very strong axiomatic assumptions on preferences, i.e., [completeness, transitivity, continuity, and independence](https://en.wikipedia.org/wiki/Von_Neumann%E2%80%93Morgenstern_utility_theorem#The_axioms) (or variations thereof).  I think this theory strongly informs the results in Appendix N since the authors claim "Sparse comparisons: Methods relying on transitive relationships, such as Elo and B-T, may struggle when comparison data is limited".  If the preferences are sparse, this violates the completeness requirements to identify a scoring-function and could very clearly explain why ELO and Bradley-Terry do poorly in this setting.  I remark that extensions such as [TrueSkill](https://proceedings.neurips.cc/paper_files/paper/2006/file/f44ee263952e65b3610b8ba51229d1f9-Paper.pdf) added "indifference" (tied rankings) to ELO and this may mitigate concerns that ELO and Bradley-Terry do not perform well with tied rankings.